# Reproducibility report: Towards Visually Explaining Variational Autoencoders

**Daisy van den Berg**
daisymvdberg@gmail.com

**Arne Meijs**
Arneb.meijs@gmail.com

**Renée Oldenkamp**
Renee.oldenkamp@gmail.com

**Mitchell Verhaar**
Mitchell.verhaar@gmail.com

## Reproducibility Summary

### Scope of Reproducibility

The paper by Liu et al. [7] claims to develop a new technique that is capable of visually explaining Variational Autoencoders (VAEs). Additionally, these explanation maps can support simple models to get state-of-the-art performance in anomaly detection and localization tasks. Another claim they make is that using these attention maps as trainable constraints leads to improved latent space disentanglement [7]. The validity of these claims will be tested by reproducing the reported experiments and comparing the outcomes with the ones of Liu et al [7].

### Methodology

To reproduce the experiments, where available, the original code provided by the authors is used. If the parameterization is not reported in the paper, the default parameters are applied. For the majority of the experiments however, no code is provided by the authors. To reproduce the experimental setups described code is sourced from other github-repositories and compared to the description in the original work. The experiments are run on a GPU-node provided by Surfsara [1].

### Results

Overall, the qualitative results attained in this reproduction study are comparable to the results given in the original paper. Showing that the attention maps highlight the anomalies in the images. However, the quantitative results do not match the original paper, as they score lower on both the AUROC and IOU metric for the anomaly detection. Also, the reconstruction of the AD-FactorVAE is not successful, thus no results for this part are obtained.

### What was easy

Running the authors code is relatively easy, the provided README elaborately explains the commands which allow you to train or test a model. Expanding the models to other architectures is relatively easy.

### What was difficult

The paper provides little information about some of the experiments and models. Also, some of the architectures shown in the paper and supplemental information are incorrect. Implementing the missing FactorVAE, Ad-FactorVAE and the disentanglement metric proved difficult and time consuming.

### Communication with original authors

The original authors replied within a couple of days to an mail containing questions. Their reply provided helpful knowledge and helped clear up some questions concerning the model architectures, completeness of the code and rectified faulty information in the paper.

---

[1]https://userinfo.surfsara.nl/

# 1 Introduction

The application of algorithms in many safety-critical and consumer-focused [2, 5, 4] areas implies an ethical responsibility to be able to prove the algorithms' fairness. Much research in the field of deeplearning is focused on ways to visualize high influencing feature regions that motivate an algorithms decision in the form of attention maps [4, 7, 10]. The drive to explain motivated Liu et al. [7] to implement a method of visualizing VAE attention through a gradient-based attention map generation method. With their method they claim to be able to explain deep generative models like VAE by highlighting crucial input feature regions on an attention map. On top of that, they show that their attention maps can be utilized to identify anomalies within images. Through qualitative and quantitative testing on three datasets they report strong performance in the fields of anomaly detection. Finally, they claim attention maps can be used as a constraint during training to improve latent space disentanglement. Again through reporting quantitative and qualitative results state-of-the-art performance is reported. This study will attempt to reproduce the findings shown by Liu et al. [7] testing the validity of the above mentioned claims.

The paper written by Liu et al. [7] is well cited, and the methods described are copied, expanded upon and compared to many novel approaches [8, 12, 14, 15, 17, 8]. The paper *Towards Visually Explaining Variational Autoencoders* by Liu et al. [7] introduced a new technique to explain VAEs and other generative models. In doing so it has inspired and influenced many consequent studies and thus testing the reproducability of this er is a meaningful contribution to the scientical community.

# 2 Scope of reproducibility

Much effort has been expended into explaining deep classification/categorization models. Liu et al [7], however, note that the methods proposed cannot be trivially extended to deep generative models such as Variational Autoencoders (VAE). In their study, Liu et al. [7] visualize VAE attention maps using a gradient-based method called GradCAM [11]. Liu et al. [7] claim the ability to intuitively explain deep generative models by visualizing the separated latent features using their attention maps. Additionally, the authors introduce a novel anomaly detection method. They propose to use their explanation generation method to provide cues on anomaly locations. The intuition behind this is that latent space representations of anomalous data should be anomalous too. Thus generating a visual explanation based on this anomalous latent feature should provide information that allows for localization of the particular anomaly. Finally, the authors propose yet another novel method, with as goal to enforce learning a disentangled latent space. They declare that using their visual attention maps as formulated disentanglement constraints, the resulting attention disentanglement learning objective provides superior disentanglement and performance when compared to existing studies.

The main focus of this study lies on the reproduction of anomaly detection results. This has many practical yet crucial applications, making explainability of the utmost importance [13] [18]. The validity of the claims made by the authors in the original paper will thus be tested. The claims that will be evaluated are the following:

- Attention map visualisations generated from an one-class VAE trained on the MNIST dataset [16], should provide an intuitive model explanation.

- Attention maps generated from an one-class VAE trained on the UCSD ped1 dataset [6] should highlight anomalies and result in comparable AUROC scores compared to performance reported in the original document.

- A VAE tested and trained on the MVTec-AD dataset [1] should highlight anomalies and generate higher AUROC and IOU scores for anomaly detection compared with other state-of-the-art anomaly detection architectures.

- An AD-FactorVAE should outperform a FactorVAE resulting in a higher disentanglement metric [3] score when trained on the dSprites dataset [9].

# 3 Methodology

As a base for the reproduction the authors code which is available on Github is used. For comparison purposes the pre-trained model for the MNIST dataset [16] shared by the authors is downloaded. Since the provided code is incomplete, some minor adjustments are applied to the code to extract latent-feature-specific attention maps as described in the original paper. To train and test the anomaly detection capabilities of a VAE on the UCSD Ped1 and MVTec-AD dataset, additional adjustments are made to the model architecture in accordance with the supplemental documentation[2],

---

[2]https://openaccess.thecvf.com/content_CVPR_2020/supplemental/Liu_Towards_Visually_Explaining_CVPR_2020_supplemental.pdf

which the authors referred to through email correspondence. Additionally, some quantitative metrics needed to be implemented to evaluate the results. This is not present and thus the original code is extended to calculate the pixel-level segmentation AUROC score. With the AUROC score thresholds could be obtained to binarize the resulting attention maps as described in the original paper. Additional code to compare these binarized maps to their true labels using an IOU scoring metric is made. Finally, an attempt at producing code for comparison between a FactorVAE and an AD-FactorVAE ismade. This includes both qualitative and quantitative testing of their latent feature disentanglement as showcased in the original paper.

## 3.1 Model descriptions

To generate visual attention maps for VAE, first, an input is encoded into a latent vector $z$ by the encoder, a convolutional neural network. Applying the reparameterization trick allows the latent space to be in the form of a multivariate gaussian distribution while still enabling backpropagation. Next, this latent space representation is fed into a decoder which attempts to reconstruct the input image. For all VAEs the training goal is to lower the reconstruction error as well as to organize its latent vector $z$ in a multivariate normal distribution of $\mathcal{N}(0, I)$. For the FactorVAE this is expanded upon with the total correlation, an untractable loss variable that is approximated by a discriminator trained alongside the VAE [3]. On top of that, the AD-FactorVAE further expands on the FactorVAEs learning goal by adding the attention disentanglement loss in the form of a constraint [7].

The four different VAEs are implemented for testing purposes. All four of the VAEs have a 32-dimensional latent space, but each one has a different encoder/decoder architecture to accommodate for the differences in the dataset used. For the MVTec-AD dataset [1] for example, a pre-trained ResNet18 with its last two layer omitted, is connected to two learned linear layers who altogether function as the encoder. The details for all of the decoder/encoder architectures can be found in the appendix. However, the layers in the encoder and decoder of the MVTec-AD architecture had to be slightly adjusted for the output dimension to be suitable to the different layers.

To generate attention maps, the elements $z_i$ of the latent vector $z$, are backpropagated to an earlier convolutional layer. This creates an attention map $M^i$ of variable granularity depending on which convolutional layer is targeted. Every $z_i$ has a corresponding attention map $M_i$. The general attention map $M$ is generated by averaging over of the separate attention maps $\frac{1}{D} \sum_i^D M^i$.

## 3.2 Datasets

Four datasets are used: MNIST [16], UCSD Pedestrian [6], Dsprites [9] and MVTec-AD [1]. The MNIST dataset [3] contains 60000 training images and 10000 testing images of 28x28 pixels. The provided split is also used during training, no test-set is used since no quantitative performance analysis is performed.

The UCSD_Ped1 dataset [4] consists of 2 separate sets of samples. Each sample is a video of around 200 frame filming a road with pedestrians, bicycles and cars passing by. It contains 34 training samples and 36 testing samples with 40 irregular events; the occurrence of vehicles such as cars or bicycles. 20% of the test samples is used as a validation set and 80% is used as test set. As for pre-processing all images are resized to 100x100.

The MVTec-AD dataset [5] is used for anomaly detection in textures or objects. It contains 5354 high-resolution color images unevenly distributed over 15 different classes (10 objects and 5 textures). All classes come with a predefined test- and train-set of which the 20% of the train-set is used as the validation-set. The train-set exclusively contains pictures that contain no defect or anomaly, while the test set contains a variety of abnormalities in the textures or objects, such as cracks or prints. All images are resized to 224 x 224 pixels.

Lastly, the dSprites data set [6] comprises 2D-shapes made from six ground truth independent latent factors. These factors are color (white), shape (square, ellipse, heart), scale (6 values in [0.5, 1]), orientation (40 values in $[0, 2\pi]$), x_coordinate and y_coordinate of a sprite. All permutations of these latent factors are present only once in the data set, which makes up for a total of 737,280 images. Since tests are not performed yet there is not test-train split to report.

## 3.3 Hyperparameters

The values of the hyperparameters concerning learning rate, batch size, and latent feature vector size are reported in the supplemental material provided by the original authors. No specifics surrounding the train time in the form of epochs

---

[3]http://yann.lecun.com/exdb/mnist/

[4]http://www.svcl.ucsd.edu/projects/anomaly/dataset.htm

[5]https://www.mvtec.com/company/research/datasets/mvtec-ad

[6]https://github.com/deepmind/dsprites-dataset

are given so for the MNIST experiment, where code is provided, tests are performed at the default settings. For the other datasets the number of epochs necessary to train the models needs to be estimated. This is achieved by running the code with multiple different numbers of epochs, in the range of 100 to 1000 with steps of 100. The best number of epochs is chosen based on the qualitative results. The optimizer used to train the model is the Adam optimizer implemented by Pytorch using the default parameters [7]. The exact values of the hyperparameters can be seen in table 1.

| | Supplemental Material | | | | |
| --- | --- | --- | --- | --- | --- |
| | Learning rate | Batch size | latent features | Epochs | Total number of parameters |
| MNIST | 0.001 | 128 | 32 | 100 | 13216193 |
| USCD Ped1 | 0.0001 | 32 | 32 | 300 | 77243073 |
| MVTec AD | 0.0001 | 8 | 32 | 300 | 72803363 |
| Dsprites (VAE) | 0.0001 | 64 | 32 | 70000 | |
| Dsprites (Discriminator) | 0.0001 | 64 | - | 70000 | |

Table 1: Hyperparameters for each model belonging to one of the four datasets.

## 3.4 Experimental setup

In total, three different experiments are executed to reproduce the first two claims of the paper. The first experiment qualitatively evaluates anomaly detection performance of a VAE on the MNIST dataset. The VAE model is trained on images of handwritten ones (digit class) and tested on images of other digit classes. This results in highlighted regions on the attention maps on the anomalous regions. This is repeated but now the model is trained using threes. The second experiment includes the qualitative and quantitative analysis of the anomaly detection with the UCSD Ped1 dataset. This dataset provides a training and test set used to obtain the attention maps showing the anomalies. For the quantitative evaluation an ablation study is done. For this, three different AUROC scores are calculated where backpropagation needed to construct the attention maps is varied with the three different convolutional layers in the encoder. The third experiment is the qualitative and quantitative evaluation of the anomaly detection on the MVTec-AD dataset. This is done for ten [8] categories. Analysis is performed by qualitative means with attention maps, and quantitatively through AUROC- and an IOU score comparison.

For reproduction purposes both the FactorVAE and the AD-FactorVAE need to be coded to train on the dsprites dataset. For the ad-FactorVAE a gradient-based method to generate latent feature specific attention maps on which it can base the disentanglement loss constraints. On top of that the disentanglement metric described by Kim et al. [3] needs to be coupled to the models. both models will be trained on the dsprites dataset and evaluated using this metric.

The experiments are run on a gpu node provided by the surfsara lisa environment [9]. These nodes contain a single GPU containing 12 cores, as well as a single CPU core. The code can be found in the github repository [10].

The experiments that are run have not been considerably taxing as far as CPU/GPU hours go. However, for the consequent tests the vaesability of the programs on a regular computer is less likely. The FactorVAE, for example, cannot run on a laptop as the demand for RAM exceeds 12GB. Tabel 2 shows the GPU hours that the three different experiments took to complete.

| | Epochs | GPU hours |
| --- | --- | --- |
| MNIST | 100 | 00:11:14 |
| USCD Ped1 | 300 | 01:22:51 |
| MVTec AD | 300 | 08:05:46 |
| Dsprites | - | - |

Table 2: GPU hours for different runs

# 4 Results

Overall, the qualitative results attained in this reproduction study are comparable to the results given in the original paper. Showing that the attention maps highlight the anomalies in the images. However, the quantitative results do not

---

[7]https://pytorch.org/cppdocs/api/classtorch$_{11}$optim$_{11a}$dam.html

[8]Carpet, Leather, Tile, Wood, Cable, Capsule, Hazelnut, Metal Nut, Pill, Transistor

[9]https://userinfo.surfsara.nl/

[10]https://github.com/reneedecoolste/FACT-AI

match the original paper, as they score lower on both the AUROC and IOU metric for the anomaly detection. Also, the reconstruction of the AD-FactorVAE is not successful, thus no results for this part are obtained.

## 4.1 Result MNIST

The reproduction of the experimental results reported by the authors tasked with a qualitative anomaly detection task is successfully reproduced. It is possible to reconstruct the qualitative results depicted in figure 4 in the original paper. This can be done by either training a VAE with the default settings of the provided code or testing the pre-trained model. A side note that should be made is that roughly 10-15% of the results are not as convincingly intuitive as the figure posted in the original study suggests. The percentage of less convincing attention maps is quantified through counting the number of less intuitive attention maps in small samples. The shown reproductions are thus made by randomly selecting the inputs and corresponding explanations displayed. As shown in figure 1 the pre-trained models resulted in identical results compared to a newly trained model.

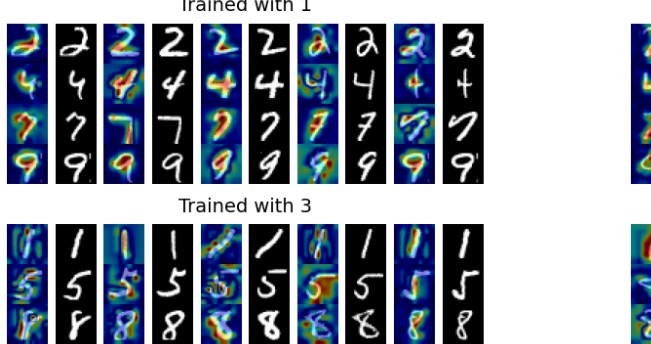 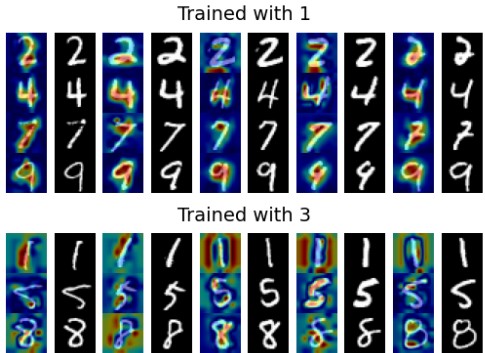

Figure 1: Reproduction of figure 4 of the original study using a random selection of samples. On the left are results of self-trained model and on the right those of the model provided by the authors. The top 4 rows containing attention maps of a VAE trained using 1's while the bottom rows show those of a model trained on 3's. Most of the interest zones shown depict intuitive anomalous zones, however some are less intuitive.

## 4.2 Result UCSD Ped1

The results obtained when reproducing the anomaly detection for the UCSD Ped1 dataset are shown in figure 2. The attention maps can be used to show the anomalies in the images. The first column shows the real images, all containing anomalies, represented by a car, a cyclist and a wheelchair. As can be seen, the anomalies are highlighted in the attention maps shown as red regions. Despite not being anomalies all pedestrians are highlighted by the attention maps too as shown in figure 2 and even in anomaly free situations as seen in figure 3. This pattern is seen uniformly throughout the results generated in the experiments. The figures showcased in the original study displayed attention maps containing more desirable highlighted anomaly regions.

|            | Conv1 | Conv2 | Conv3 |
|------------|-------|-------|-------|
| Reproduced | 0.58  | 0.58  | 058   |
| Liu et al. | 0.89  | 0.92  | 0.91  |

Table 3: AUROC scores for UCSD Ped1 dataset where the attention maps are constructed by backpropagating to the three different convolutional layers. Result compared to the results of Liu et al. [7]

## 4.3 Results MVTec-AD

The reproduction of the qualitative results of the MVTec-AD dataset can be seen in figure 4. As shown, the images are correct in the highlighting anomalies and resemble the qualitative results given in the paper. However, these results are

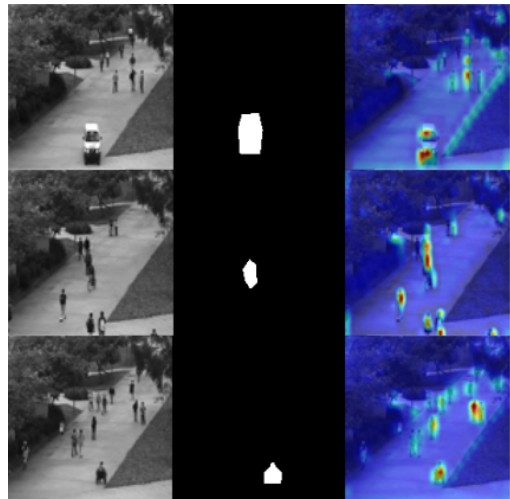

Figure 2: Anomaly detection of the USCD Ped1 dataset. L-R: Original image, the ground truth (the location of the anomaly), and the attention maps. In the first row the anomaly is a car, in the second row a cyclist and in the third row a wheelchair. These anomalies are found as shown in the attention map but also some pedestrians are classified as anomalies.

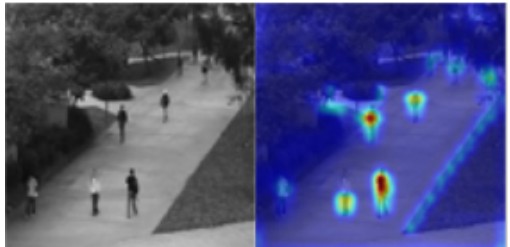

Figure 3: Anomaly detection of an image that has no anomalies, although the attention map shows some pedestrians being classified as anomalies.

cherry picked and most of the other resulting attention maps did not display the correct anomaly detection. By choosing a random subset of ten images of all ten categories about 70% of the results did not correctly highlight the anomalies.

The quantitative results are shown in table 4, the AUROC score and the IOU scores for ten categories are adjoined to the results of the original paper. Both the AUROC and the IOU scores of the reproduction resulted in lower scores than the original paper scores for all ten categories.

| | Reproduced | Liu. et al. |
|---|---|---|
| Carpet | 0.39 | **0.78** |
| | 0.007 | **0.10** |
| Leather | 0.47 | **0.95** |
| | 0.045 | **0.24** |
| Tile | 0.56 | **0.80** |
| | 0.079 | **0.23** |
| Wood | 0.50 | **0.77** |
| | 0.017 | **0.14** |
| Cable | 0.27 | **0.90** |
| | 0.006 | **0.18** |
| Capsule | 0.16 | **0.74** |
| | 0.011 | **0.11** |
| Hazelnut | 0.82 | **0.98** |
| | 0.058 | **0.44** |
| Metal Nut | 0.78 | **0.94** |
| | 0.242 | **0.49** |
| Pill | 0.79 | **0.83** |
| | 0.084 | **0.18** |
| Transistor | 0.40 | **0.93** |
| | 0.008 | **0.30** |

Table 4: Quantitative results for pixel level segmentation on 10 categories from MVTec-AD dataset. The AUROC score is notated at the top row, the IOU is notated at the bottom row. The scores reproduced scores can be compared to the scores of Liu et al. [7]

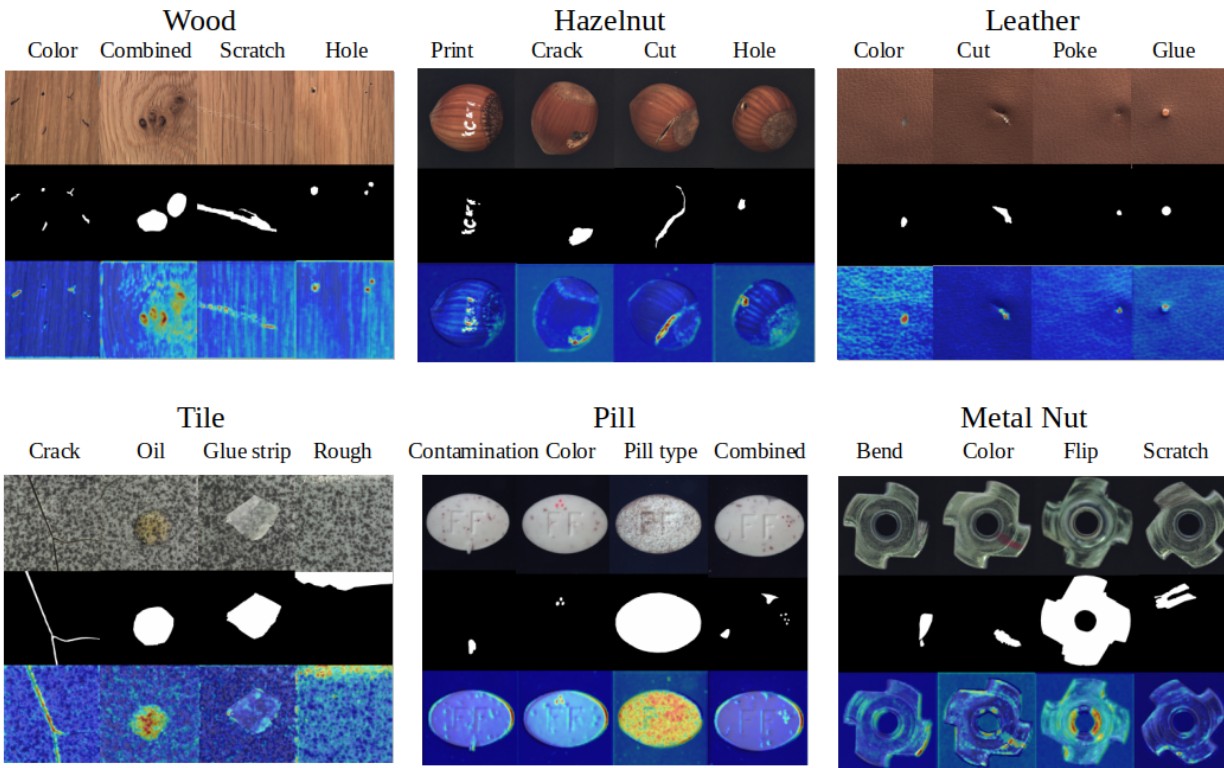

Figure 4: Qualitative results for anomaly detection for 6 of the categories in the MVTed-AD dataset, showing four different types of defects for each category

### 4.4 Results Latent disentanglement

Although a FactorVAE is successfully trained on the dSprites datasets, it could not successfully be testes as this implementation is not completed in time. Unfortunately it is not possible to reproduce the results for the disentanglement tests, the task of implementing and extending a FactorVAE as well as connecting a vaguely described measurement metric proved outside the capabilities within the allotted time.

## 5 Discussion

It is possible to reproduce results for the MNIST dataset confirming the first claim, providing intuitively model explanation. Second, it is possible to reproduce some qualitative results for the UCSD Ped1 dataset supporting the claim that it highlights anomalies. However, the quantitative results do not support this claim as the score are low, in conclusion it not possible to validate this claim based on the reproduced results. The same applies to the third claim, for which qualitative results on the MVTec-AD dataset support the claim, but quantitative results score too low compared to the reported . The last claim, stating the attention maps could improve latent space disentanglement, could not be validated based on the provided information, as the code needed to test this is not successfully reproduced. Multiple factors may have influenced the results in a positive or negative way.

### 5.1 What was easy

When reproducing this paper, some factors are experienced as easy. First, the code that the author provided is easy to set up and generated instant results for the MNIST dataset, as the model weights and architectures are shared.

Second, the supplementary materials of the paper paved the way for the architectures of the encoders and decoders in both the UCSD Ped1 and MVTec-AD data set implementations of the paper. This greatly simplified the implementation issues one might come across when reproducing the research.

Third, the contact with the authors of the paper is experienced as relatively simple and very helpful as they answered to the email quite fast and with useful information, among which the attention of the existing supplemental material.

## 5.2 What was difficult

Since the code provided by the authors is limited to the explanation generation on the MNIST dataset, the rest has to be distilled from the paper and other independent reproductions on github. This is experienced as relatively hard for some parts of this reproductions study. Some factors, considered as difficult, may have badly affected the approach and the results.

In terms of the provided implementation a lot of documentation is lacking. The code provided in the GitHub is poorly if not at all documented and several design choices are not listed at all. An example of this is the splitting of the data sets in training and validation parts. The amount of images per part is not defined at all. However, leaving such an important trait up to potential reproducers of the research may lead to inaccurate results.

Another issue is that it is difficult to recreate the authors testing environment as a result of the sporadic documentation for the hyperparameters. The disentanglement performance metric is not provided in code and neither is it explained in the paper. It has to be entirely distilled from another paper as there is also no code available for it in the repository. These factors increases the difficulty to reproduce the original test environment, which may influence the results as well.

Dealing with the data sets, apart from the MNIST data set since an implementation is already provided, caused some difficulties as well. Importing the data sets into the model required each of the data sets to have its' own defined class, because each of them is structured and represented differently. Especially with the lacking of any code or opportunity to link the data to the model in a simplified way, this turned into a troublesome task.

The MVTec-AD data set is accompanied with its' own separate set of issues in this paper. The authors describe that the categories within this data set are augmented to create 10000 training images for each data set. They do so by transforming the images that already exist in each category by a random amount of degrees within the range of $-30.0$ and $30.0$. Furthermore, in able to add even more diversity to the data the authors also mirror some of the objects in the images. However, there is no clearly defined size of each of these augmentations. Additionally, no explanation of this mirroring operation is provided at all. The mirroring is described as being used *when the object permits it*, which is a vague metric to measure with. These concerns cause problems when trying to reproduce the data set augmentations. The authors of this paper do refer to the original research that accompanied this data set, but this research does not provide the necessary explanation as well. This part may have influenced the low AUROC and IOU scores for the MVTec-AD datasets, as the training data might not be correct.

Even though the architecture of the VAE model used with the MVTed-AD is provided in the supplemental material this still caused some difficulties as the exact architecture did not work when implemented. It had to be adjusted in order to work.

Overall the reproduction of this paper is experienced as quite difficult because of insufficient knowledge about pytorch, resulting in considerable time that needed to be put in understanding the available code and finding ways to extend it.

Overall, it is considered that due to time limits the experiments could not be performed as good as possible. First, it was not possible to run multiple seeds, from which the results would benefit. Second, it was not possible to implement the AD-FactorVAE. Third, less hyperparameters could be tested than preferred.

To summarize, the factors have influenced the approach and results in a negative way. A next reproduction study will benefit if, preferably all, factors are taken into account.

## 5.3 Communication with original authors

The authors of the article are very responsive and helpful in clearing our questions about their article. They responded within reasonable time on the email send to them, and provided proper insights in their set-up, and referring to their supplemental material. They also stated that figure 2 of their paper, which explains their element-wise attention generation with a VAE, needed to be adjusted, after we notified that the Relu in this figure is not used in the available code.

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

 # 6 Appendix

| Network | Layer | Output Dimensions |
|---|---|---|
| Encoder | Conv 2D, $4 \times 4$, 64,2,1 | $14 \times 14 \times 64$ |
| | ReLU | $14 \times 14 \times 64$ |
| | Conv 2D, $4 \times 4$, 128,2,1 | $7 \times 7 \times 128$ |
| | ReLU | $7 \times 7 \times 128$ |
| | Flatten | 6272 |
| | Linear | 1024 |
| | ReLU | 1024 |
| | Linear | 32 |
| Decoder | Linear | 1024 |
| | ReLU | 1024 |
| | Linear | 6272 |
| | ReLU | 6272 |
| | Unflatten | $7 \times 7 \times 128$ |
| | ReLU | $7 \times 7 \times 128$ |
| | ConvTr 2D, $4 \times 4$, 64,2,1 | $14 \times 14 \times 64$ |
| | ReLU | $14 \times 14 \times 64$ |
| | ConvTr 2D, $4 \times 4$, 1,2,1 | $28 \times 28 \times 1$ |
| | Sigmoid | $28 \times 28 \times 1$ |

(a) Model used for anomaly detection in the MNIST dataset.

| Network | Layer | Output Dimensions |
|---|---|---|
| Encoder | Conv 2D, $4 \times 4$, 64,2,1 | $50 \times 50 \times 64$ |
| | ReLU | $50 \times 50 \times 64$ |
| | Conv 2D, $4 \times 4$, 128,2,1 | $25 \times 25 \times 128$ |
| | ReLU | $25 \times 25 \times 128$ |
| | Conv 2D, $4 \times 4$, 256,2,1 | $12 \times 12 \times 256$ |
| | ReLU | $12 \times 12 \times 256$ |
| | Flatten | 36864 |
| | Linear | 1024 |
| | ReLU | 1024 |
| | Linear | 32 |
| Decoder | Linear | 1024 |
| | ReLU | 1024 |
| | Linear | 36864 |
| | ReLU | 36864 |
| | Unflatten | $256 \times 12 \times 12$ |
| | ReLU | $256 \times 12 \times 12$ |
| | ConvTr 2D, $5 \times 5$, 128,2,1 | $25 \times 25 \times 128$ |
| | ReLU | $25 \times 25 \times 128$ |
| | ConvTr 2D, $4 \times 4$, 64,2,1 | $50 \times 50 \times 64$ |
| | ReLU | $50 \times 50 \times 64$ |
| | ConvTr 2D, $4 \times 4$, 1,2,1 | $100 \times 100 \times 1$ |
| | Sigmoid | $100 \times 100 \times 1$ |

(b) Model used for anomaly detection in the UCSD Ped1 dataset.

| Network | Layer | Output Dimensions |
|---|---|---|
| | Input Image | $64 \times 64$ |
| Encoder | Conv 2D, $4 \times 4$, 32,2,1 | $32 \times 32 \times 32$ |
| | ReLU | $32 \times 32 \times 32$ |
| | Conv 2D, $4 \times 4$, 32,2,1 | $16 \times 16 \times 32$ |
| | ReLU | $16 \times 16 \times 32$ |
| | Conv 2D, $4 \times 4$, 64,2,1 | $8 \times 8 \times 64$ |
| | ReLU | $8 \times 8 \times 64$ |
| | Conv 2D, $4 \times 4$, 64,2,1 | $4 \times 4 \times 64$ |
| | ReLU | $4 \times 4 \times 64$ |
| | Conv 2D, $4 \times 4$, 128,1,1 | $1 \times 1 \times 128$ |
| | ReLU | $1 \times 1 \times 128$ |
| | Conv 2D, $1 \times 1$, 32,1,0 | 32 |
| | Conv 2D, $1 \times 1$, 32,1,0 | 32 |
| | Input | $\mathbb{R}^{32}$ |
| Decoder | Conv 2D, $1 \times 1$, 128,1,0 | 128 |
| | ReLU | $1 \times 1 \times 128$ |
| | ConvTr 2D, $4 \times 4$, 64,1,0 | $4 \times 4 \times 64$ |
| | ReLU | $4 \times 4 \times 64$ |
| | ConvTr 2D, $4 \times 4$, 64,2,1 | $8 \times 8 \times 64$ |
| | ReLU | $8 \times 8 \times 64$ |
| | ConvTr 2D, $4 \times 4$, 32,2,1 | $16 \times 16 \times 32$ |
| | ReLU | $16 \times 16 \times 32$ |
| | ConvTr 2D, $4 \times 4$, 32,2,1 | $32 \times 32 \times 32$ |
| | ReLU | $32 \times 32 \times 32$ |
| | ConvTr 2D, $4 \times 4$, 1,2,1 | $64 \times 64 \times 1$ |

(c) Model used for the disentanglement of the latent space of the Dsprites dataset.

| Network | Layer | Output Dimensions |
|---|---|---|
| Encoder | Resnet18(w/o last 2 layers) | $8 \times 8 \times 512$ |
| | Linear | 1024 |
| | Linear | 32 |
| Decoder | Linear | 1024 |
| | Linear | $1024 \times 4 \times 4$ |
| | ConvTr 2D, $4 \times 4$, 512,2,1 | $8 \times 8 \times 512$ |
| | BatchNorm | $8 \times 8 \times 512$ |
| | ReLU | $8 \times 8 \times 512$ |
| | ConvTr 2D, $4 \times 4$, 256,2,1 | $16 \times 16 \times 256$ |
| | BatchNorm | $16 \times 16 \times 256$ |
| | ReLU | $16 \times 16 \times 256$ |
| | ConvTr 2D, $4 \times 4$, 128,2,1 | $32 \times 32 \times 128$ |
| | BatchNorm | $32 \times 32 \times 128$ |
| | ReLU | $32 \times 32 \times 128$ |
| | ConvTr 2D, $4 \times 4$, 64,2,1 | $64 \times 64 \times 64$ |
| | BatchNorm | $64 \times 64 \times 64$ |
| | ReLU | $64 \times 64 \times 64$ |
| | ConvTr 2D, $4 \times 4$, 32,2,1 | $128 \times 128 \times 32$ |
| | BatchNorm | $128 \times 128 \times 32$ |
| | ReLU | $128 \times 128 \times 32$ |
| | ConvTr 2D, $4 \times 4$, 3,2,1 | $256 \times 256 \times 3$ |
| | Sigmoid | $256 \times 256 \times 3$ |

(d) Model used for anomaly detection in the MVTec-AD dataset. Note the fact that a flatten() operation is missing between the ResNet18 module and the consequent linear modules. This module is added because the output of ResNet18 would not have a suitable dimensionality for the consequent linear layers. Also, a linear layer with output dimensions 16384 followed by a Unflatten(1024, 4, 4) is added instead of the linear layer with output 1024x4x4 as this was not possible.

Figure 5: Here the four models, four different implementations of the variational autoencoder, can be seen.

