# OpenReview forum: "Reproducing towards visually explaining variational autoencoders"
_ML_Reproducibility_Challenge/2020 — Reject_

### Official Review · AnonReviewer2 · 2021-03-01
**Visually explaining auto encoders  - reproducibility report**

**Rating:** 4
**Confidence:** 3

**Review:**

The authors have attempted to reproduce the results of the article "Towards Visually Explaining Variational Autoencoders" by Liu et. al. (CVPR 2020). They have presented a summary report which has all the required elements.

The authors have made a significant effort to reproduce the results of the original paper. They mentioned that they obtained some of the code from the authors and sourced the rest from other github-repositories. They also report having contacted the authors to clarify information in the original paper.

The report is reasonably well organized though the explanation of the original paper in the Introduction could use improvement. They could include some more details about the methodology so as to enable understanding it without reading the original paper.

Some of the other concerns I have with the report are as follows:

1. They report that the original paper's authors' code was incomplete but do not clarify what additions/adjustments they had to make to the code. They have not provided their modified codebase.
2. The authors' primary focus is to reproduce the anomaly detection results and report that they were unable to reproduce all the results of the original paper. They also report that their results for the quantitative metrics do not match those of the original paper. However, the authors have not tried to explain why  The code for quantitative metrics used to evaluate the results was written by the authors themselves. Is it possible that there were errors/differences in the evaluation code? There is no mention of any correspondence between them and the original paper's authors to discuss the differences in the results. Their final conclusions do not include any analysis of the differences in the results.
3. It is impressive that they have reported the training time for each of the data sets that they have tested on but they have not due diligence on the hyperparameter sweep. The only hyperparameter they have performed a search on is the number of epochs. They have used the default values  (as specified by the original paper's authors in their supplementary material) for the other hyperparameters.
4. Though the reproduction of the results of latent disentanglement using the dSprites dataset is included in the scope of the reproducibility they have not performed these tests.

In summary, though the authors have made a significant effort and shown the difficulties in reproducing the results of the original paper,  I believe that this report could use a lot of improvement.


**Familiar With The Original Paper:**

I have read the original paper

**Reproducibility Summary:**

Report has summary

---

### Official Review · AnonReviewer1 · 2021-03-02
**Well written, but not experiments not exhaustive.**

**Rating:** 7
**Confidence:** 3

**Review:**

The original paper was difficult to reproduce because of the lack of certain parts of the code and poor documentation, however, the authors did a good job in reproducing and verifying various claims. As per the results, the quantitative values reported here are lower than expected. The paper is very well written. What is lacking is a thorough hyperparameter exploration.


**Familiar With The Original Paper:**

I have read the original paper

**Reproducibility Summary:**

Report has summary

---

### Official Review · AnonReviewer3 · 2021-03-03
**Review...**

**Rating:** 7
**Confidence:** 4

**Review:**

The authors provide a very nice summary and concrete that is being elaborated in the document. They cover fully the reproducibility of the paper in terms of scope where they reused the original author's code. It would be interesting to perform a hyperparameter search in order to verify the claims of the authors and maybe find new hyperparameters not experimented with the authors.

In general, the report is well written and has a clear structure so I recommend it for being accepted.

**Familiar With The Original Paper:**

I have not read the original paper

**Reproducibility Summary:**

Report has summary

---

### Decision · Program_Chairs · 2021-03-31

**Decision:**

Reject

**Comment:**

While the authors have made a significant effort and shown the difficulties in reproducing the results of the original paper, there is a lot more work to be done to do a solid reproduction and the report.